## Perspective

trauma; stress; self-care; resiliency; interoception; public health; mental health; intervention; well-being; task-sharing

**Corresponding author:**
Linda Grabbe;
Email: lgrabbe@emory.edu

# The Community Resiliency Model, an interoceptive awareness tool to support population mental wellness

Linda Grabbe ⬤, Ingrid M. Duva and William C. Nicholson

Nell Hodgson Woodruff School of Nursing, Emory University, Atlanta, GA, USA

## Abstract

The objective of this article is to describe the Community Resiliency Model (CRM)®, a sensory-focused, self-care modality for mental well-being in diverse communities, and CRM's emerging evidence base and neurobiological underpinnings as a task-sharing intervention. Frieden's Health Impact Pyramid (HIP) is used as a lens for mental healthcare interventions and their public health impact, with CRM examples. CRM, a sensory awareness model for self-care and mental well-being in acute and chronic stress states, is supported by neurobiological theory and a growing evidence base. CRM can address mental wellness needs at multiple levels of the HIP and matches the task-sharing concept to increase access to mental health resources globally. CRM has the potential for making a significant population mental health impact as an easily disseminated, mental health, self-care modality; it may be taught by trained professionals, lay persons, and community members. CRM carries task-sharing to a new level: scalable and sustainable, those who learn CRM can share the wellness skills informally with persons in their social networks. CRM may alleviate mental distress and reduce stigma, as well as serve a preventive function for populations facing environmental, political, and social threats.

## Impact statement

Somatic mental wellness techniques to bolster resiliency are simple to teach and can be used immediately. The Community Resiliency Model (CRM)®, a unique self-care set of skills derived from somatic psychotherapy, is a tool to promote mental health globally and can be adapted for diverse populations. Learners gain a compassionate understanding of their own responses to stress and trauma, and learn fundamental skills to better withstand stress, using the body itself as a resource for strength. Lay workers and community members may become certified to formally teach CRM and those they teach can informally teach CRM to others, with a multiplicative effect, and taking task-sharing to a new level. We report here on CRM's burgeoning research base supporting significant improvement in mental well-being measures for populations across varying contexts. This is an effort to draw global attention to CRM among persons who work in the service of others. CRM has utility both as a self-care modality to reduce burnout potential and as an efficient wellness intervention with the potential for widespread community dissemination in low-resource areas.

The global need for simple, accessible, evidence-based mental health interventions existed prior to the COVID-19 pandemic, with only a small fraction of those needing help ever receiving care (Campion, 2018). The pandemic exacerbated the crisis, engendering stress, anxiety, depression, insomnia, denial, anger, and fear (Torales et al., 2020), exposing defects in already fragile and inadequate systems of care (Auerbach and Miller, 2020; Gribble et al., 2022). Large-scale disasters, social turmoil, political conflict, climate change, population displacement, community violence, and structural racism all place population mental health at risk and underscore the need for mental health interventions that are effective, low-dose, transferrable, and sustainable (WHO, 2022). Task-sharing interventions that incorporate laypersons in mental healthcare delivery are of particular interest as scalable strategies that can greatly increase access to care.

Task-sharing is a pragmatic approach to increasing health service capacity by redistributing work tasks among healthcare providers and community members (Orkin et al., 2021). Task-sharing interventions must be scalable, that is, they must have sustained cost-effectiveness and impact when they are fully implemented or broadly delivered. Healthcare providers delegate or share responsibilities with community members or healthcare workers who have less training or lower skill levels, but are trained to provide specific services. This creates capacity and resources in settings where there are unmet needs or large treatment gaps. Healey et al. (2022) conducted an economic evaluation of a scaled-up, evidence-based, task-sharing mental health intervention called the Friendship Bench in Zimbabwe. Their analysis found this task-sharing intervention to

be a cost-effective way to treat depression and other common mental disorders in low- and middle-income countries (Healey et al., 2022). The burden of common mental disorders is also great in high-income countries like the United States where there are healthcare inequities and where a large treatment gap exists for the medically underserved and rural populations. Task-sharing is a promising solution, if interventions are evidence-based and scalable.

The Community Resiliency Model (CRM)® is a preventive and trauma-sensitive mental wellness self-care intervention. While existing task-sharing mental health interventions appear to focus on cognitive or psychological approaches to trauma (Ryan et al., 2021), these approaches often miss the biological underpinnings of human resiliency and wellness. With its roots in trauma psychotherapy, CRM is a non-cognitive, "bottom-up," wellness program – a significant departure from "top-down" cognitive models (Grabbe and Miller-Karas, 2018). Developed at the Trauma Resource Institute by Elaine Miller-Karas, CRM may be valuable as a widely acceptable, lay-delivered modality for community mental health prevention that can also be deployed in emergency situations. The purpose of this article is to explain the CRM, a quickly taught set of emotion regulation skills based on simple sensory awareness. Thomas Frieden's Health Impact Pyramid (HIP) (Frieden, 2010), a public health framework for categorizing healthcare interventions, serves to illustrate the impact of mental health interventions and where CRM can fit in the schema to support global mental health needs.

## Brief overview of the Community Resiliency Model and its emerging evidence

CRM is a variant of present-moment awareness (mindfulness), emphasizing attunement to one's own internal (interceptive) and external (exteroceptive) cues for regulation of autonomic responses to stress and trauma (Miller-Karas, 2015, 2023). CRM did not emerge from mindfulness practices; rather, it originated as a simple self-care modality based on a rich body-focused psychotherapy tradition (Levine, 2010; Miller-Karas, 2015; Ogden, 2015). Body-based psychotherapists use "bottom-up" (somatic) rather than a "top-down" (cognitive) techniques for clients to learn to self-regulate using their own body prior to trauma processing. Self-regulation capacity is developed by teaching the client to attend to body sensations and to differentiate between sensations of distress and those of well-being, strength, and resilience, before and if any trauma is recounted. Body-based approaches in psychotherapy are of particular value to trauma survivors (Van der Kolk, 2015).

CRM is part of a paradigm shift occurring in healthcare, education, and social service realms toward strengths-based, trauma-informed, and resiliency-focused wellness practices (Advocates for Trauma-Informed Policies, n.d.). Antecedents to CRM occurred in natural and man-made disaster settings in the 2000s; under emergency conditions with no time for psychotherapy, responders schooled in Peter Levine's Somatic Experiencing psychotherapy model (Levine, 1997) taught disaster survivors how to access internal psychophysiological resources for emotion regulation. After these disasters, those victims experienced lower than expected rates of post-traumatic stress disorder (PTSD); further, emergency responders themselves were able to use the skills as prevention against the impact of secondary stress (Parker et al., 2008; Leitch et al., 2009; Leitch and Miller-Karas, 2009). CRM grew out of these expert-driven solutions and natural experiments and has since been developed and used as an emergency intervention (e.g., Florida

Pulse Nightclub Shooting; Las Vegas Shooting at Route 91 Harvest Music Festival; Paradise, California Wildfire) (Trauma Resource Institute, n.d.).

CRM's cornerstone concept, the "Resilient Zone" (RZ) (Figure 1a), normalizes and de-stigmatizes human responses to stress and trauma. When we are in the RZ (our bandwidth for stress tolerance), we function adequately and have the emotional balance that affords clear, intentional, goal-directed action and thoughts, even in the presence of mild stress perturbations (Miller-Karas, 2015, 2023). When we are overwhelmed by threat or negative emotion, we are pushed into a hyper-aroused state (too much sympathetic discharge in the autonomic nervous system) or into a hypo-aroused state (too much parasympathetic) (Figure 1b). It feels uncomfortable to be outside of the RZ, and our attempts to feel better account for the many unhealthy behaviors we engage in (e.g., substance use and aggression) to deal with emotional tension.

The goals of CRM are 1) to widen the RZ over time and 2) to use skills to return to the RZ when knocked out by threat or strong emotion. Stress responses are explained in biological terms, and CRM's self-regulation skills are themselves biological tools: somatic awareness of interoceptive (inside the body) and exteroceptive (via the five senses) sensations (Miller-Karas, 2015, 2023). CRM skills are described elsewhere (Grabbe et al., 2021), but the "tracking" skill, awareness of sensation (interoception or exteroception), is fundamental to all CRM skills. Healthcare workers who have learned CRM's interoceptive and exteroceptive techniques (e.g., touching their skin or clothing and noting body sensations) report using them to maintain their composure during tense, chaotic, and crisis situations (Grabbe et al., 2020; Grabbe et al., 2021). The intentional use of CRM reduces or interrupts attention to unpleasant sensations, thoughts, or behavioral responses by reorienting focus to other less salient or neutral sensations, thus altering our stress responses.

CRM's burgeoning research base includes its use for the long-term, chronic stress of marginalized groups in the United States (Grabbe et al., 2020; Freeman et al., 2021), and globally, in post-Ebola Sierra Leone communities (Aréchiga et al., 2023) and with Rwandan genocide survivors (Habimana et al., 2021). Research including two randomized controlled trials of single 1- or 3-h trainings (virtual or in-person) has demonstrated significantly improved mental well-being and reduced stress symptoms in persons in high-stress occupations (Leitch et al., 2009; Grabbe et al., 2020, 2021; Duva et al., 2022). Other variables that demonstrated significant improvements over time included somatic symptoms, depression, anxiety, and PTSD.

## Possible neurobiological mechanisms underlying the effectiveness of CRM

A central feature of CRM skills is physiological awareness or interoception. Interoception involves sensing, integrating, interpreting, and regulating body signals, and is central to body homeostasis and to the regulation and control of emotions (Chen et al., 2021). These sensory processes are on-going in the body, providing internal status updates within and outside of our awareness and allowing us to identify and adjust for discomforts such as temperature, hunger, fatigue, or pain. Most sources of stress originate from the self, others, and environmental threat. Any single one or combination of these three elements augment the stress burden (Ottaviani, 2018). The more difficult or psychologically complex the cause of the stress, the more overwhelmed we become, and the more we are likely to leave the RZ or get "stuck" outside the RZ in a dysregulated

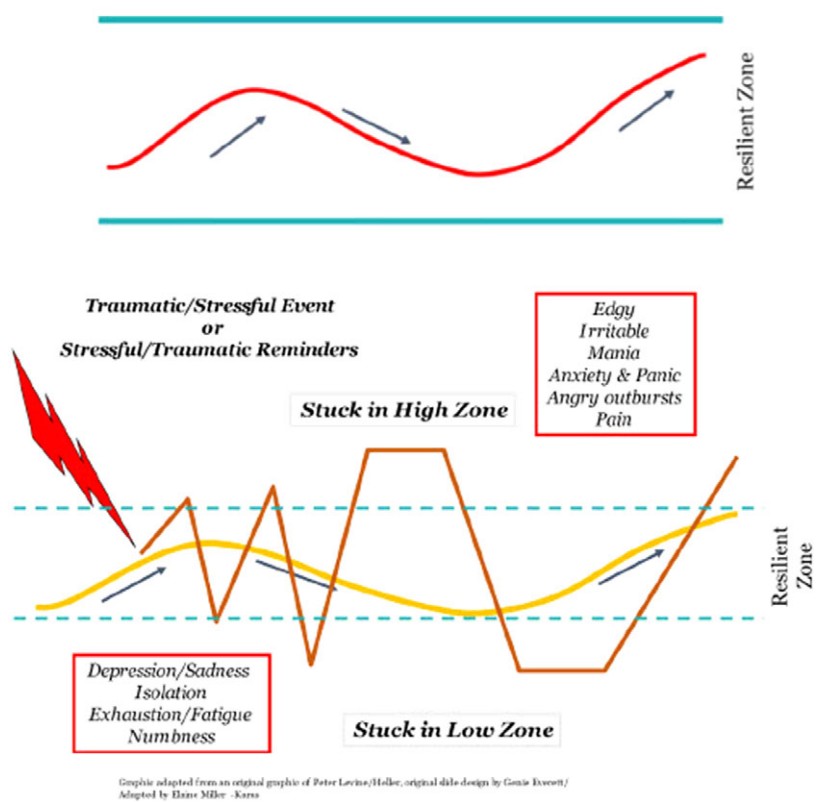

**Figure 1.** The resilient zone: (a) regulation and (b) dysregulation.

state. Ultimately, if this dysregulated physiological stress cycle persists, there is an increased risk of chronic physical and mental disorders (Anda et al., 2006).

When we are in the RZ, we have executive brain resources to override emotionally driven, impulsive behaviors; however, under stress (if we are knocked out of the RZ), this override is blunted or lost. Under acute stress conditions, the brain attempts to 1) identify the source of stress, 2) respond to the stressor (e.g., fight/flight), 3) neutralize its effect on the body, and 4) learn from the experience (identifying cause or context). In acute stress, higher-level executive brain functions are sacrificed in favor of more immediately responsive, emotionally driven responses; top-down control is reduced so that metabolic energy can be shunted to the stress response to protect us via bottom-up control (Gagnon and Wagner, 2016). Under acute conditions of unpredictable stress exposure, this redirection of metabolic resources away from executive domains is a protective mechanism that allows for optimization of threat neutralization.

Although the default loss of executive override is protective in acute stress, it can also be deleterious, and even to think clearly, we may need to counteract this protective response. To do so, even momentary physiologic awareness may interrupt our default protective mechanisms. Interventions that counteract the loss of executive override in severe stress reactions can be of supreme value to maintaining mental balance and well-being. CRM's intentional body awareness seems to provide access to self-regulation via

bottom-up pathways, even under trauma conditions; this affords immediate integration of bottom-up and top-down activity at the time of stress (Weng et al., 2021). Under intense stress, cognitive interventions that rely solely on intact executive pathways to help regulate stress may not be as useful as bottom-up modalities like CRM that focus on sensing changes in physiology.

Focusing on body sensations involves sensory activation and coordination between autonomic, interoceptive, exteroceptive, and proprioceptive pathways (Rohe and Zeise, n.d.). Integration of these sensory pathways modulates segments of the limbic system where information-processing and body regulation occur. Internal representations about self are processed and co-coordinated in these regions (Carvalho and Damasio, 2021; Chen et al., 2021). This coordination appears to produce many aspects of resiliency: self-control, attachment, social engagement, sense of self, well-being, empathy, and cooperation (Gogolla, 2017; Koban et al., 2021). In effect, CRM may rely on changes in the insular pathways dedicated to processing body state and self-representation, to influence attention mediated by higher-ordered cognitive centers.

CRM appears to provide immediate access to psychological rebalancing by directing attention away from inherently self-centered, stress-based thoughts, toward greater coherence with the way that we perceive our relationship in the "self-other-environment" system (Feldman, 2020). In light of CRM's growing research base, its potential to support population mental wellness can be further clarified via a population health conceptual framework.

### Conceptual framework: The Health Impact Pyramid

Frieden's five-tiered HIP was created to classify health interventions into five levels according to their impact on the health of a society (Frieden, 2010). Figure 2a illustrates the levels of interventions required to keep a society healthy. At the base of the pyramid are interventions that have the greatest possible public health impact, and at the top of the pyramid are interventions that require 1:1 encounters that have a strong individual effect, but negligible population impact. Frieden contends that the maximum possible sustained public benefit regarding a given health problem entails interventions implemented at multiple levels. We use the framework here to classify mental health interventions, providing examples of practical CRM applications (Figure 2b).

### Mental healthcare through a HIP lens

Tier 1 interventions tackle the social determinants of mental health. These efforts include population-level primary prevention to promote mental well-being in the population and reduce the risk factors for mental illness, addiction, violence, and illness. Such change would include the availability of mental health resources to all citizens, and the prevention and treatment of critical upstream factors of population mental health, for example, adverse childhood experiences, which affect the majority of populations (Felitti et al., 1998; Sederer, 2016; Teicher and Samson, 2016).

Tier 2 interventions require structural or systemic changes such that healthy choices are the default course of action. Efforts to develop the social and emotional health of children in schools,

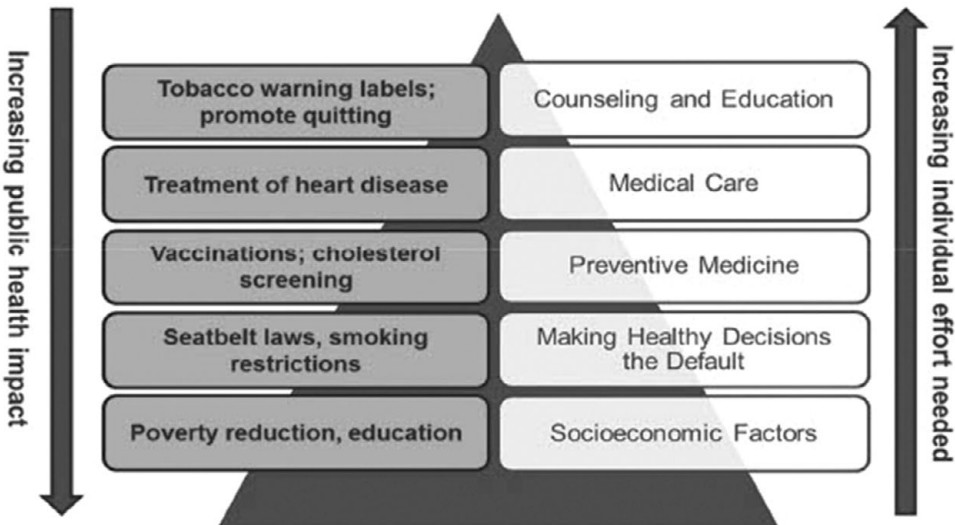

a) The Health Impact Pyramid. Framework for public health action: the health impact pyramid. Frieden TR. Am J Public Health 2010;100:590–5.

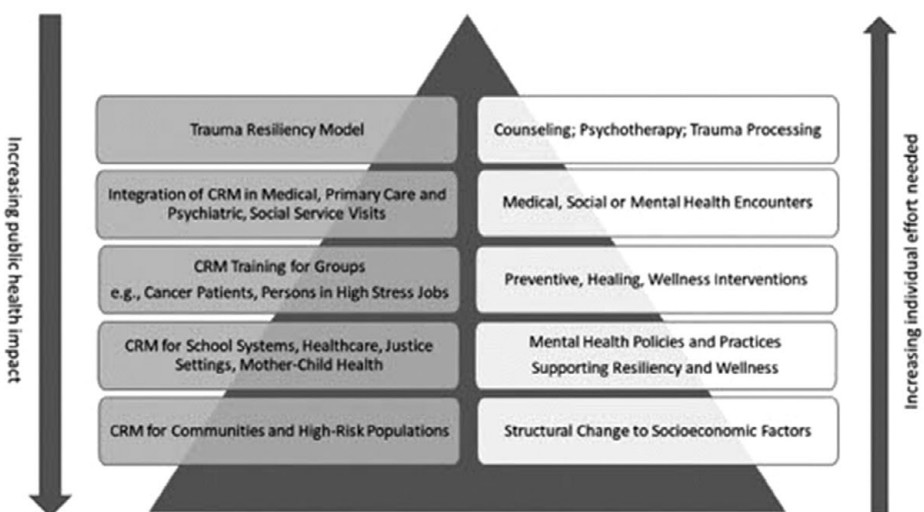

b) The Health Impact Pyramid with categories of mental health interventions (right) and examples of CRM and TRM (left)

**Figure 2.** A public health perspective: a) health interventions and b) mental health interventions.

and community interventions to increase trauma- and resiliency-informed programs across public and private sectors are current moves in the right direction (Advocates for Trauma-Informed Policies, n.d.; Building a Collaborative Georgia, n.d.). Tiers 1 and 2 require considerable economic and political effort, and for mental health, this drive must be culturally and contextually relevant to counteract stigma and the many variables that lead to adverse mental health outcomes. Interventions that incorporate local knowledge and culture, use existing resources, and ensure equitable distribution of mental health resources are ideal (Gil-Rivas et al., 2019).

Tier 3 interventions include brief or one-time interventions that confer long-range protective benefits without further clinical follow-up. Although Frieden describes Tier 3 interventions as individually focused (Frieden, 2010), preventative interventions for depression, suicide, or substance use can reach large numbers of people simultaneously via virtual or in-person programs via classes, hotlines, podcasts, and apps that raise awareness and prompt individual or community action. Although Frieden does not mention task-sharing, this model for dissemination of preventative programs fits naturally here. Health and mental well-being can be promoted when content is taught by both behavioral health providers and members of local communities who are trained to teach the material. Further, it is local experts who are best able to adapt a given program to the needs of their own communities.

Tier 4 interventions are individually focused clinical encounters to manage mental health problems and prescribe medications, in primary care and psychiatric settings. Tier 5 interventions are 1:1 counseling, psychotherapy, and psychoeducational encounters. Tier 4 and 5 encounters may have potential for the greatest impact on individuals, but their population effect is low. Access at these tiers is limited by inadequate insurance coverage for mental healthcare, and sparse resources, especially in rural and underserved areas, and in low- and middle-income countries. Outcomes of these intervention are further affected by client adherence to treatment and willingness to make behavioral changes. These are the costliest interventions and are time- and labor-intensive for patients and providers alike. The majority of U.S. health spending occurs at the top two layers of the pyramid (McCullough et al., 2020), highlighting the failure to address mental health needs of the U.S. population.

## CRM through the HIP lens

CRM has applications at multiple HIP tiers. At Tiers 1 and 2, a global movement and free curriculum for Social, Emotional and Ethical (SEE) Learning that incorporates CRM for school children has been launched in dozens of countries by Emory University (SEE Learning, n.d.). SEE Learning was being put into place in all Ukrainian schools prior to the Russian invasion. Since then, CRM training has been available to all Ukrainians online in a humanitarian effort by the Trauma Resource Institute, where CRM was developed (Trauma Resource Institute, n.d.). CRM may be taught at an institutional level, as it is for all students in Wake County, North Carolina[1] and as it is for sexual assault police interview training in the State of Washington.[2]

------

[1] https://www.wcpss.net/Page/47968.
[2] https://www.traumaresourceinstitute.com/blog/2020/12/30/ambassador-of-the-year-jen-wallace.

At Tier 3, CRM is a long-term, preventative intervention because certified teachers can train large numbers of persons simultaneously in person or virtually and those learners can practice the skills as they wish to with no clinical intervention for the rest of their days (Trauma Resource Institute, n.d.; Duva et al., 2022). In addition, because CRM is a task-sharing modality, local community members can be trained as certified teachers, reaching far greater numbers of people, and adapting the model to suit local needs (Miller-Karas, 2015, 2023). It is also important to consider an even greater potential social impact: any person who grasps CRM concepts and skills can share them with persons in their own social networks in a one-to-one fashion (as a "CRM Guide"), creating an exponential mental wellness effect in our populations, at no cost whatsoever. The free iChill app can be used if desired for reinforcement and will promote confidence in sharing the wellness skills.

At Tiers 4 and 5, CRM can be easily integrated into primary care, psychiatric, or social service visits. Care providers who learn CRM concepts and skills can teach them one-to-one as a CRM Guide, and do not need formal training to do so. In any healthcare encounter, providers can integrate strength-focused conversational CRM questions (accompanied by active listening), such as "What is helping you get through this right now?" to bolster clients' resilience. CRM is explained in a free smartphone app ("iChill") or at http://www.ichillapp.com/ for reinforcement and to share with others. The concepts and skills may be blended with many psychotherapy models. A sister to CRM and a trauma processing tool for mental health professionals, the Trauma Resiliency Model, was also developed by Miller-Karas (2015, 2023).

## Discussion

Population mental health includes the promotion of mental well-being and the prevention and treatment of mental and substance use disorders. To manage mental health distress and work toward well-being for themselves and their communities, people need skills that will reduce their suffering and cultivate their resilience. The HIP underscores the need for equitable, multi-faceted mental health interventions to reach the greatest numbers of people. Task-sharing mental health interventions fit into the lower tiers of the HIP, assuring the greatest possible impact.

Task-sharing interventions are used globally in an attempt to address the treatment gap in mental healthcare, and while evaluations support their scalability, other results have been mixed. HIV-positive patients with depression in Zimbabwe received lay-counselor-delivered sessions of a problem-solving therapy and had promising improvement on depression scores (Abas et al., 2018), but an intervention with lay workers conducting counseling sessions for women with perinatal depression in South Africa demonstrated no impact (Lund et al., 2020). Both of these interventions involved cognitive content/strategies. A complementary skills-based interoception component might have altered the response to these interventions. Standard mental health interventions generally use cognitive approaches, but a mix of cognitive and body-based programs are ideal for trauma-related mental health problems (Corrigan and Hull, 2015; Van der Kolk, 2015).

Mental health first aid (MHFA) is a globally disseminated task-sharing training for members of the public to act as the first responders for mental health issues in their communities, but a recent literature review of MHFA found no effects on the helpfulness of trainees' actions or on recipient mental health (Forthal et al., 2022). MHFA includes assessing, listening, and offering support; a

skill-building addition of CRM would teach biologically based coping skills on the spot. For some, the skills might be sufficient to alleviate symptoms; for others, CRM might serve as a bridge until professional behavioral healthcare can be accessed. In the U.S. State of Georgia, CRM is being initiated as a "Bridge to Therapy" approach for children with behavioral problems as they await assessment and treatment that may be months away.[3] Because common responses to stress and trauma are seen as biological in nature, rather than as moral or character weakness, CRM removes some of the stigma of having a mental health challenge and receiving formal behavioral healthcare.

To put a CRM task-sharing intervention into practice, program planners may use an implementation science framework. The Barriers and Facilitators in Implementation of Task-Sharing Mental Health Interventions (Le et al., 2022) focuses on low-income countries, but high-income countries where rural and underserved urban populations experience extreme healthcare inequities, are essentially in this category. We hope this introduction to CRM will elicit further investigation of the intervention as a scalable task-sharing modality to tackle the mental health treatment gap in low-resource settings. To expand CRM's evidence base, further research is warranted in 1) emotion regulation biomarkers (cortisol levels, electroencephalography, and heart rate variability); 2) impact on specific mental health problems (anxiety and depression) or behaviors (substance use, violence, and incarceration); 3) positive outcome parameters (well-being, emotion regulation, resiliency, and pro-social elements such as teamwork, empathy, self- and other-compassion, and communication); and 4) community or public health outcomes, real-world scalability, and resolution of treatment gaps.

## Conclusion

The World Health Organization's call for low-dose, transferrable, and sustainable mental health interventions to deal with the burgeoning global mental health crisis highlights the need for models of mental healthcare which can 1) reduce vulnerability to mental health problems (primary prevention), 2) decrease the sequelae of these problems (secondary prevention), and 3) promote healing (tertiary prevention). CRM is part of shift toward a strengths-based, biologically focused resiliency perspective for population mental health, and a front-end self-care wellness strategy to help people experiencing mental distress. The HIP illustrates that the greatest impact on population health requires interventions at the lower tiers of the pyramid. CRM is a strategy that incorporates task-sharing at two levels: non-behavioral healthcare providers can become certified to deliver CRM concepts and skills to their communities and organizations; in addition, persons they teach can informally share the model with persons in their own networks, augmenting the impact on global mental well-being.

**Open peer review.** To view the open peer review materials for this article, please visit http://doi.org/10.1017/gmh.2023.27.

**Author contribution.** L.G. and I.M.D. conceived the study and determined the methodology. All authors contributed to the writing and organization of the manuscript and reviewed the final manuscript before submitting for publication.

**Financial support.** No funding was received for this work.

---

[3]https://www.youtube.com/watch?v=O7rlO3Rg2pw.

**Competing interest.** None of the authors has any competing interest to declare. L.G. and I.M.D. receive speaking fees for presentations at conferences on the topic of this manuscript.

**Free CRM recordings.** 1. Emory Nursing Experience (with continuing education credit): https://ce.emorynursingexperience.com/courses/cultivating-our-best-selves-in-response-to-covid-19.
2. Trauma Resource Institute (Resources): https://www.youtube.com/watch?v=mX3KTqFUA-E.

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
