## [Reviewer Report]

Dear Drs. Chibanda, Bass, and Belkin,

Our manuscript “The Community Resiliency Model (CRM), an Interoceptive Awareness Tool to Support Population Mental Wellness” introduces an exciting response to ongoing mental health needs of individuals and the demands for emotional capacity in communities around the world. CRM can be a one-time, self-care, mental-wellness modality that lends itself naturally to task-sharing. CRM’s burgeoning research base and neurobiological underpinnings are explained in our article, and we use Frieden’s Health Impact Pyramid to illustrate how this intervention can fit at multiple tiers of health systems for mental health promotion, prevention, and treatment. 

Mental health providers, healthcare organizations, and community leaders are looking for efficient, cost-effective ways to build resiliency, vibrancy and justice within their systems. CRM is a set of easily learned concepts and somatic awareness skills that use the body itself to cope with the stresses of everyday life, the cumulative trauma of strife/racism/oppression, and the immediate impact of natural and man-made disasters. Recent research, including randomized controlled trials, demonstrate that CRM’s self-regulation techniques can help learners reduce traumatic stress symptoms and increase sense of well-being. We present a brief summary of these findings, as well as CRM research with Tutsi genocide survivors and post-Ebola community members in Sierra Leone. CRM is a progressive solution to the need for a self-care, task-sharing mental health intervention. Please note that although the three authors are Certified CRM teachers, we have no conflicts of interest with the Trauma Resource Institute, where CRM was developed.

Sincerely,

Linda Grabbe, PhD, FNP-BC, PMHNP-BC, Professor Emeritus

Nell Hodgson Woodruff School of Nursing, Emory University

---

## [Reviewer Report]

Thank you for providing me with this significant opportunity to review your manuscript. This is critical perspective research that will benefit the sciences.

I agree to publication with minor changes. There’s no need to return to me if you change these little suggestions.

1. Add reference on page.“3” in the second paragraph .." even in the presence of mild stress perturbations(??). 

2. Put the number of the figure in parentheses: ex: Figure(2a) to avoid confusing readers.

3. Page(10): use footnotes on the youtube link "https://www.youtube.com/watch?v=O7rlO3Rg2pw)." to avoid confusion 

4. On page (7), there is a double point in the subtitle, but on page (8), there is not. Kindly adjust.

5. Last, adjust the words,“ ”discussion, “ ”conclusion, " or other subtitles either in justify, centered, or left alignment. Make your subtitles in similar alignment

---

## [Reviewer Report]

Thank you for the opportunity to review this article. The authors make a compelling argument for the need for interventions such as CRM and provide theoretical and research support for its effectiveness. However, there are a few areas for improvement that could strengthen the article:

1. Please provide a brief discussion of the term “scalable” as it relates to Healey’s work.

2. Please clearly state that CRM as a trauma-sensitive (TS) model in the first sentence of the second paragraph.

3. It would be helpful to connect the discussion of the TS model more clearly with the Health Impact Pyramid by highlighting why both are important for promoting health and well-being.

4. There is a missing paragraph indent on page 4 that needs to be corrected.

5. Please explain how techniques, used in CRM, such as touching one’s skin or clothing, are interoceptive and how this is helpful.

6. The intentional use of body awareness in CRM reduces attention to unpleasant sensations and alters the stress response. How so (moving aways form unpleasant sensations)?

7. Interventions that counteract the loss of executive override in severe stress reactions can be of supreme value to maintaining mental balance and well-being. Even during an actual trauma? When is the override needed and not needed?

8. To better integrate the discussion of CRM as a TS model with the Health Impact Pyramid, it would be helpful to mention the potential benefits of the TS model in the HIP section.

---

## [Reviewer Report]

Dear Dr. Chibanda

Thank you for this opportunity to submit the revision of our manuscript, “The Community Resiliency Model (CRM), a Task-Sharing Interoceptive Awareness Tool to Support Population Mental Wellness.” We hope we have adequately addressed the reviewer comments and will be glad to add any further details. We are including the edited MS and the edited MS with highlights where we made changes. Linda Grabbe

---

## [Reviewer Report]

This paper is well described and has the potential to help the world community to know more about the intention they were expressing.